# The abilities in dog pain sign recognition as assessed by presenting seventeen listed dog behavioural signs and three case descriptions to dog owners and non-dog owners

Silvia M. A. Gardeweg , Dionne E. Picard, Ineke R. van Herwijnen *

Animals in Science and Society, Animal Behaviour Group, Faculty Veterinary Medicine, Utrecht University, Utrecht, The Netherlands

* i.r.vanherwijnen@uu.nl

## Abstract

To investigate abilities and differences in dog pain sign recognition, we assessed these recognition skills in 530 dog owners and 117 non-dog owners through an online questionnaire. We asked participants to score the likeliness of pain relevance for seventeen dog behavioural signs and in three dog behavioural cases. When assessing the seventeen behavioural signs, the signs of 'change in personality', 'hesitant paw lifting', 'fluctuating mood' and 'reduced play' were scored at higher pain likeliness scores than 'air sniffing', 'nose licking' and 'yawning'. The behaviours of 'turning the head or body away' and 'freezing' were scored at higher pain likeliness scores by non-dog owners than dog owners. The cases twice regarded behaviours of a dog with a painful condition and once without such a condition. One painful condition came with overt pain signs, related to movement ability once and the other painful condition came with subtle pain signs, such as shadowing family members and restlessness at night. We found that participants rated the likeliness of pain significantly higher in the case describing overt dog pain signs, related to movement ability. Dog owners' ratings for this case were slightly higher than non-dog owners' ratings. Yet, for the case describing subtle dog pain signs, such as shadowing and restlessness, no differences were found between dog owners and non-dog owners. This may indicate that dog owners recognise subtle dog pain signs with less ease. Possibly, dog owners recognise signs such as 'turning head or body away' and 'freezing' (more so) as a stress/fear sign, than as a possible pain sign. We argue that education on dog behaviour may benefit from addressing how behavioural signs may also, or alternatively, be indicative of pain. Our finding that owners of dogs that experienced a painful event, scored a higher likeliness of certain possible pain signs, indicates that experience matters and this underpins how education on dog behaviour could possibly benefit animal welfare through addressing subtle pain signs in dogs.

**Data availability statement:** All relevant data are within the manuscript and its Supporting information files.

**Funding:** The author(s) received no specific funding for this work.

**Competing interests:** The authors have declared that no competing interests exist.

## Introduction

In view of the continuing growth in companion animal ownership of about 3% per year, approximately 46% of the European households accommodate at least one companion animal [1]. After cats, living in 26% of households, dogs follow closely, living in 25% of households [2]. With such high numbers of dogs living with humans, it is of interest to know how dog ownership affects human well-being [3–7] and how dogs are influenced by living with humans [8–11]. A clear risk for both dogs and their owners is biting risk [12–14], as seen in dog bites causing 350.000 US emergency department treatments in 2018 [13] and 1.6 million treatments between 2005–2009 [14]. A dog's biting may stem from several causes and motivational states, among which fear motivation and pain experiences [15–18]. Risk of biting may increase if people living with or around dogs fail to adequately observe and interpret a dog's body language [19,20]. Body language of a dog seems relevant, as often dog bites occur when people interact with (familiar) dogs, as was seen in familiar dogs causing a biting incident in 66% (N = 320 of 484) of self-identified UK dog bite victim cases [21]. Adequate recognition of warning signs from dog body language may prevent or stop risky interactions from occurring.

Both dog owners and non-dog owners may find it challenging to observe and interpret dog body language correctly [22]. In recent years, studies have provided new insights on how people may use a dog's facial expression to recognise certain emotions [23,24]. However, studies on *subtle* dog behaviour recognition seem less available to date, particularly with regard to recognising the emotion of pain. One study on dog emotion recognition, that included pain as a secondary emotion, focused on facial versus bodily expression channels [25]. The study indicated that dog ownership related to higher recognition of certain emotions than non-dog ownership. Yet, dog ownership related negatively to recognising 'sad facial expression' and no significant relationship was found for dog ownership and recognition of pain [25].

The lack of studies exploring the ability of dog owners and non-dog owners in interpreting a dog's subtle pain signs, could be regarded a scientific gap. These subtle signs in a dog's behaviour might provide early warning signs for dog aggression, allowing for mitigation of the previously mentioned dog biting risks, if pain is causal to a dog's aggression. When in pain, dogs may behave unpredictably and react to stimuli that they usually would not react to with aggression [26]. Thus, the recognition of subtle behavioural signs might be of value to human health and wellbeing. Also, dog welfare might benefit, as recognition of subtle pain signs could lead to more timely pain intervention [21]. A dog's pain experience may at first, or only, be reflected in subtle signs and missing these signs may prolong and/or worsen a dog's suffering [26]. In dogs and other animals, pain signs are often subtle. In a narrative review, examples of subtle signs for chronic pain included a more withdrawn demeanour, reduced sociability and play, reluctance or refusal to perform activities such as jumping into a car [27]. As a consequence of the subtleness of some pain signs, it may be difficult to recognise pain signs in dogs. Yet, the recognition and correct interpretation of subtle behavioural signs might be necessary to prevent aggression and (further) suffering of a dog in pain.

People do recognise that dogs can experience pain, as seen in over 75% of participants in a survey of 438 dog and cat owners reporting that dogs can experience pain [28]. To recognise pain, more than half of the participants studied body posture, facial expression, speed of movement and vocalisation [28]. If humans with and without dogs recognise a dog's pain experience and look at certain signs, this offers opportunities to teach them also the subtle signs of a dog's pain. For instance, educational tools could be developed to increase dog pain sign awareness. To be able to develop such educational tools, an advanced insight into dog (subtle) pain sign recognition is needed and a recent article by Mills et al. [29] offers a basis to assess this recognition. Mills et al. lists signs that are not yet widely recognised as possible pain signs, such as lip licking, air sniffing, reduced playing and increased grooming [29]. As a first step towards improving recognition of subtle pain signs in dogs, we studied skills in dog owners and non-dog owners regarding the identification of a dog's behavioural signs being possibly indicative of pain. We did so, based on three cases and the signs listed in Mills et al. [29] as 'common signals and cues to discomfort', such as pain. Following the information provided in the second paragraph (above), that both dog owners and non-dog owners find it challenging to observe and interpret dog body language correctly [21], we hypothesized that dog owners do not recognise subtle dog pain signs more so than non-dog owners. Based on studies indicating that a person's own painful experiences affect perception of pain signs [30,31], we secondly hypothesized that participants with a personal painful experience recognise subtle dog pain signs more so than those without such experience. Thirdly, we tested in dog owners only, if previous experience with a dog undergoing a painful experience, comes with improved recognition of subtle dog pain signs, based on an assumption that dog owners may have received education by veterinarians on subtle pain signs at the time of the painful experience of their dog [32].

## Methods

### Questionnaire development

An online questionnaire, composed of four sections, was created in Dutch, using Qualtrics® Experience Management. The first section consisted of six artificial intelligence generated images, three of these depicting dogs showing postures and behaviours indicative for pain (praying position, hunched back, lifted paw) according to current literature [29,33], and three displaying dogs performing routine behaviours and postures (play bow, hunting position and waiting position). The images were presented in mixed order to participants, asking to rate the likelihood of the displayed behaviours being indicative for pain on a numerical scale from 0 to 4 (0 = very unlikely, 1 = moderately unlikely, 2 = neutral, 3 = moderately likely, 4 = very likely). In the present study we did not use the study results from these images, as we aim to study these further in a separate study and focused the present article on pain sign recognition based on the descriptions of pain signs and cases. The second and third section of the questionnaire were central to this manuscript. In the second section, we tested participant's ability to recognise subtle dog pain signs based on a list of 'common signals and cues to discomfort' listed in a study by Mills et al. [29]. To facilitate understanding for participants without scientific background, we used the term 'pain', not the term 'discomfort'. This allowed us to limit the number of questions presented to participants, as we only needed to include items on pain, and not also on discomfort. We discuss this choice in the Discussion section of our article. The seventeen signs listed in the study by Mills et al. [29] were presented to the participants in a matrix. We asked the participants to evaluate each sign for its likelihood to be considered as a pain sign, using the numerical scale introduced in the first section and indicated above. A definition of pain was not provided, as not to influence the participants' initial perception and interpretation of the seventeen signs (and the three cases).

In the third section, three separate cases of dogs were described in a vignette format. The first case, with subtle pain signs, referred to a dog suffering from panosteitis, a painful inflammatory disease [34], and the second case, with overt pain signs, pictured a dog with patella luxation, an orthopaedic condition often associated with chronic pain [29,35]. The third example described a dog which behaviour was motivated not through pain experience but through learning processes based on a prey drive after the neighbours started keeping rabbits in their garden [36,37]. The cases are described in S1 File and were created by a veterinarian and dog behavioural specialist. Pain descriptions regarded for

the first, subtle pain sign, case: increased attachment behaviour, shadowing adult family members, restlessness at night, not lying rolled up anymore, shortening the park walk; and for the second, overt pain sign, case: hopping, keeping left leg raised, less enthusiasm for park walk, lesser play with ball, opting for dog cushion not couch. Participants were instructed to rate the likelihood of six potential motivations of the dog's behaviour (fear; hormones, such as in puberty; learning processes; the dog's upbringing; pain; boredom), using the same numerical rating scale provided in the first two sections. In addition, participants were questioned to specify which of five possible reasons (0 = not a reason; 1 = a reason, with the option to select multiple behaviours) for the dog's behaviour made them opt for the motivation they rated at the highest likeliness.

The last section of the questionnaire consisted of questions regarding the participant's age and gender, possible pain experience ('Have you experienced a) a painful treatment, b) a painful disease, c) a painful accident historically or presently?'), as well as supplementary questions addressing exclusively dog owners, about their dog's age, for how long they had lived with the dog, sex and neutering status of the dog, if their dog was currently experiencing any physical (health) problems and if so which. Also, we asked if their dog underwent a painful treatment or suffer(ed) from a disease or accident in the present or past (grouped as 'painful experience'). If participants filled in that they had a dog in the past but not at present, they were asked to indicate this dog's painful experiences. Lastly, we inquired about their dog's housing situation.

## Participant recruitment and ethical considerations

A convenience sample was targeted via several social media platforms using a link providing access to the online questionnaire. The targeting included a picture of a dog and a short introductory text, describing that the study was about dog behaviour without specifying the study aim to assess pain recognition. Also, participants were informed of the anonymous collection of the data. After this, participants were asked if they wished to partake in the study and ticked a box to confirm consent if they wished to do so. Alternatively, they closed the browser if they did not wish to partake. The questionnaire was accessible between November 22nd and December 17th, 2024 to individuals aged 18 years or older.

This project was part of a larger project for which we sought and received approval for both humans and dogs via the ethical boards responsible for reviewing research projects for these species. For humans specifically, which were the population of this survey-based study, ethical approval was sought and obtained from the Science-Geosciences Ethics Review Board of Utrecht University (22nd of May, 2024; ERB Review Vet S-24.002).

## Data management and statistical analysis

We exported data from Qualtrics® to Microsoft Excel® and used IBM SPSS® Statistics version 29 for statistical analyses. For each of the above indicated seventeen behavioural signs and for each of the three cases, using the numerical scale indicated before, we expressed the indicated pain likeliness scores as mean ± S.D. for all participants and for dog owners and non-dog owners separately. For the cases we additionally present descriptive data for reasons provided by participants (overall) to select pain as a motivation in the three vignette cases.

Next, we combined scores of '0' (very unlikely), '1' (moderately unlikely) into 'not (very) likely' and we combined the scores of '3' (moderately likely) and '4' (very likely) into '(very) likely', keeping '2' (neutral) as the intermediate score. This combination was made as our interest was in knowing which signs people would likely regard as possible pain signs. For these new scores, we tested for differences with Mann-Whitney U tests a) between the dog owners and non-dog owners and b) between previous painful experiences or lack thereof in 1) participants and 2) dogs of dog-owning participants. To correct for multiple testing, we manually applied a Bonferroni correction to the P-values, by dividing a P-value of 0.05 by the number of 17 pairs tested for the pain sign comparisons, following [38]. This, as SPSS® does not automatically perform a Bonferroni correction for Mann-Whitney U tests. Thus, we regarded P-values of 0.003 as significant for the testing

of the pain signs; regarding P<0.05 as a trend. For the other comparisons, that were ≤5 in the number of comparisons, we deemed a correction not necessary and regarded P<0.05 as significant.

## Results

### Participant characteristics

A sample of 647 participants consisted of 530 dog-owners and 117 present non-dog owners. Dog owners regarded participants who presently had one or more dogs. Of the 117 non-dog owning participants, 54 had lived with a dog at some point during their lives, but presently did not own a dog. Of the 117 non-dog owning participants, 14 walked a dog, which took place for 13 of these less than weekly and for one person weekly or more often. The dogs of the dog-owning participants were mostly (55%, N=290) adult, aged between three and ten years old, with 16% (N=84) under three years old and 29% (N=156) above ten years old. Most dogs had lived with their owners for longer than a year (89%, N=471; 11%, N=59 were with their owner for a year or less). Male and female dogs were evenly spread, but dogs owned by participants were more likely to be neutered (neutered female 36%, N=192; intact female 14%, N=75; neutered male 27%, N=142; intact male 23%, N=121).

Most participants were 50–65 years old (42%, N=271; 18–35 years 24%, N=153; 35–50 years 25%, N=161; 65 years or older 9%, N=61, N=1 missing value). When looking at dog owners and non-dog owners separately, this distribution was different, with the non-dog owning sample being relatively younger. More than half were 18–35 years (53%, N=62; 35–50 years 13%, N=15; 50–65 years 27%, N=32; 65 years or older 7%, N=8). Most participants were female (88%, N=568; male 11%, N=69, other or prefer not to say 2%, N=10). When looking at dog owners and non-dog owners separately, this distribution was dissimilar, with more participants identifying with the male gender among the non-dog owners (21%, N=22; female 78%, N=91; other or prefer not to say 2%, N=2; dog owners: male: 11%, N=69; female 88%, N=568; other or prefer not to say 1%, N=10). Most participants reported to not experience pain or to experience only mild pain at the time of filling out the questionnaire (83%, N=538; average to strong 15%, N=95; prefer not to say 2%, N=14). When looking at dog owners and non-dog owners separately, this percentage was for the dog owners slightly lower: 82%, N=433; average to strong 16%, N=83; prefer not to say 3%, N=14 and for non-dog owners slightly higher: 90%, N=105; average to strong 10%, N=12.

Of the 647 participants, a total of 56% (N=363) indicated to have previously experienced a painful accident, illness or treatment (no such experience: 40%, N=261; prefer not to say 3%, N=23). Of the 530 dog owners 54% (N=286) indicated to have previously experienced a painful accident, illness or treatment (no such experience: 42%, N=222; prefer not to say 4%, N=21). Of the 117 non-dog owners 66% (N=77) indicated to have previously experienced a painful accident, illness or treatment (no such experience: 33%, N=39; prefer not to say 1%, N=1). The distribution between accident, illness and treatment is presented in S2 Table. Of the dog owning participants 55% (N=290) had previous experience with their dog's painful accident, illness and/or treatment, with the percentages per experience presented in S3 Table.

### The seventeen behavioural signs

Using the list of seventeen possible pain signs as explained in the Methods section, we assessed to which extent these were seen likely as indicative of pain by calculating means±S.D.. We first assessed this for all participants, so for both dog owners and non-dog owners together. The three behaviours with the highest mean pain likeliness scores were: 'change in personality', 'hesitant paw lifting', 'fluctuating mood' and 'reduced play' (≥3.1 on a scale from 0–4). The three behaviours with the lowest mean pain likeliness scores were: 'nose licking', 'yawning' and 'air sniffing' (≤1.8). S4 Table presents all means±S.D..

### The seventeen behavioural signs – dog ownership

Next, with Mann-Whitney U tests, we assessed how dog owners and non-dog owners might differ in attributing pain likeliness to the seventeen behavioural signs. Dog owners and non-dog owners did not differ in their recognition of possible pain signs with the exception of three behaviours. Non-dog owners gave higher likeliness scores for turning the head or

body away (z=−3.51, P<0.001) and freezing (trend: z=−2.68, P=0.007). Dog owners gave higher likeliness scores for licking surfaces (trend: z=−2.68, P=0.007; Table 1). S5 Table provides the observed frequencies per behavioural sign.

### The seventeen behavioural signs – participants' painful experiences

We assessed with Mann-Whitney U tests how a participant's own previous painful experience could come with differences in recognising possible pain signs. Participants with and without such experience did not differ in their recognition of possible pain signs with the exception of a trend found for three behaviours. The dog's changed personality (z=−2.22, P=0.027), increased blinking (z=−2.24, P=0.025) and yawning (z=−2.05, P=0.040) were indicated more often to be a (very) likely pain sign by those with previous painful experience than those without (N=23 missing values), see Table 2. S6 Table provides the observed frequencies per behavioural sign.

### The seventeen behavioural signs – dogs' painful experiences

Finally, we assessed with Mann-Whitney U tests how dog owners' previous experience with their dog suffering a painful accident, illness and/or treatment could come with differences in recognising possible pain signs. Table 3 shows how participants with such experiences indicated pain sign-likeliness more often for two behaviours, with a trend for six additional behaviours. The two behaviours regarded changed look and reduced play (P<0.001). The six behaviours regarded change in personality, coat changes, increased blinking, increased grooming, increased scratching and surface licking (P<0.05). For the behaviours of air licking/sniffing, fluctuating mood, freezing, hesitant paw lifting, lip/nose licking, turning the head or body away and yawning such differences were not found (all P>0.05). S7 Table provides the observed frequencies per behavioural sign.

### The three cases

Using the results from the interpretation of three cases describing dog behavioural changes, we assessed to which extend pain would be attributed as a motivation to the behavioural changes, in comparison to the other possible motivations by

**Table 1. Possible pain sign recognition by dog owners and non-dog owners. Mann-Whitney U tests indicated a trend for two behavioural signs and a significant difference for one sign when comparing pain likeliness scores between 530 dog owners and 117 non-dog owners (P<0.05).**

|  | Dog owners (N=530) | Non-dog owners (N=117) |
|---|---|---|
| *Turning head or body away* *(z=−3.51, P<0.001)* |  |  |
| Not (very) likely | 21% (N=111) | 7% (N=8) |
| Neutral | 27% (N=145) | 20% (N=31 |
| (Very) likely | 52% (N=274) | 67% (N=78) |
| *Freezing* *(z=−2.68, P=0.007)* |  |  |
| Not (very) likely | 23% (N=124) | 18% (N=21) |
| Neutral | 34% (N=178) | 24% (N=28) |
| (Very) likely | 43% (N=228) | 58% (N=68) |
| *Licking surfaces* *(z=−2.68, P=0.007)* |  |  |
| Not (very) likely | 25% (N=134) | 38% (N=45) |
| Neutral | 27% (N=141) | 24% (N=28) |
| (Very) likely | 48% (N=255) | 38% (N=44) |

**Table 2. Recognition of a dog's possible behavioural pain signs in participants with and without a previous painful experience.** Mann-Whitney U tests indicate differences per possible behavioural pain sign between 647 participants of which 363 indicated to have previously suffered a painful accident, illness and/or treatment and 261 indicated to have not previously suffered such an event (N = 23 missing values; P < 0.003 regarded as significant and P < 0.05 regarded as a trend).

| | Previous painful experience (N = 363) | No such previous experience (N = 261) |
|---|---|---|
| *Changed personality (z = −2.22, P = 0.027)* | | |
| Not (very) likely | 1% (N = 3) | 2% (N = 4) |
| Neutral | 7% (N = 27) | 12% (N = 32) |
| (Very) likely | 92% (N = 333) | 86% (N = 225) |
| *Increased blinking (z = −2.24, P = 0.025)* | | |
| Not (very) likely | 19% (N = 69) | 23% (N = 60) |
| Neutral | 25% (N = 92) | 31% (N = 81) |
| (Very) likely | 56% (N = 202) | 46% (N = 120) |
| *Yawning (z = −2.05, P = 0.040)* | | |
| Not (very) likely | 35% (N = 129) | 41% (N = 107) |
| Neutral | 28% (N = 101) | 31% (N = 81) |
| (Very) likely | 37% (N = 133) | 28% (N = 73) |

calculating means ± S.D. We first assessed this for all participants, so for both dog owners and non-dog owners together. We found that for the first case of a dog with subtle pain signs such as shadowing family members, the motivation of pain was scored at lower mean likeliness scores (2.4 ± 1.3, 0–4) than the second case with overt, movement ability-related pain signs (3.7 ± 0.7, 0–4), but not the third 'non-pain' case (0.7 ± 1.0, 0–4). S8 Table presents all means ± S.D.

For each case, we next asked which of five possible *reasons* made participants opt for the motivation that they rated at the highest likeliness, as mentioned above. This allowed us to assess the reasons that did ('1') or did not ('0') make them opt for a pain motivation, for all three cases and additionally for learning processes for the third case (S9 Table presents all data, also for the third, non-pain related case). For the first case of a dog with subtle pain signs, the two high percentages were for 'restlessness at night' (67%, N = 234 of 347 pain assigned motivation) and 'shortening the park walk' (65%, N = 227 of 347 pain assigned motivation). The 'restlessness at night' did not differ significantly for reason attribution between pain or not (67%, N = 234 of 347 pain assigned motivation versus 65%, N = 196 of 300 non-pain assigned motivation, P = 0.6). 'Shortening the park walk' did differ significantly (65%, N = 227 of 347 pain assigned motivation versus 29%, N = 87 of 300 non-pain assigned motivation; z = −9.13, P < 0.001). Low percentages were for 'shadowing adult family members' (32%, N = 110 of 347 pain assigned motivation) and 'increased attachment behaviour' (35%, N = 120 of 347 pain assigned motivation), with both differing significantly for reason attribution between pain or not (for shadowing adult family members: 32%, N = 110 of 347 pain assigned motivation versus 56%, N = 167 of 300 non-pain assigned motivation; z = −6.14, P < 0.001); for increased attachment behaviour: 35%, N = 120 of 347 pain assigned motivation versus 45%, N = 136 of 300 non-pain assigned motivation; z = −5.13, P < 0.001).

For the second case of a dog with overt pain signs related to movement ability, the highest two percentages of pain-indicating participants, were for the reasons of 'keeping the left hind leg raised' (90%, N = 555 of 620 pain assigned motivation versus 26%, N = 7 of 27 of non-pain assigned motivation; z = −2.50, P = 0.01) and 'hopping' (74%, N = 461 of 620 pain assigned motivation versus 37%, N = 10 of 27 of non-pain assigned motivation; z = −4.26, P < 0.001); with no percentages under 50%.

## The three cases – dog ownership

Next, we assessed with Mann-Whitney U tests how dog owners and non-dog owners might differ in recognising pain as a motivation for the cases. We found no differences between these for the first case of a dog with subtle

**Table 3. Recognition of a dog's possible behavioural pain signs in dog owners with and without a dog that previously suffered a painful experience. Mann-Whitney U tests indicate differences per possible behavioural pain sign between 530 dog owning participants of which 240 indicated experience with a dog that previously suffered a painful accident, illness and/or treatment and 290 indicated to lack such experience (P<0.003 as significant, P<0.05 as a trend).**

| | Dog with previous painful experience (N=240) | Dog without such previous experience (N=290) |
|---|---|---|
| *Change in personality (z=−1.99, P=0.046)* | | |
| Not (very) likely | 1% (N=2) | 2% (N=6) |
| Neutral | 7% (N=16) | 11% (N=31) |
| (Very) likely | 93% (N=222) | 87% (N=253) |
| *Changed look (z=−3.46, P<0.001)* | | |
| Not (very) likely | 9% (N=22) | 12% (N=35) |
| Neutral | 23% (N=56) | 36% (N=105) |
| (Very) likely | 68% (N=162) | 52% (N=150) |
| *Coat changes (z=−2.13, P=0.033)* | | |
| Not (very) likely | 5% (N=13) | 8% (N=24) |
| Neutral | 19% (N=45) | 24% (N=70) |
| (Very) likely | 76% (N=182) | 68% (N=196) |
| *Increased blinking (z=−2.14, P=0.033)* | | |
| Not (very) likely | 21% (N=49) | 23% (N=66) |
| Neutral | 21% (N=51) | 30% (N=87) |
| (Very) likely | 58% (N=140) | 47% (N=137) |
| *Increased grooming (z=−2.42, P=0.015)* | | |
| Not (very) likely | 11% (N=25) | 13% (N=39) |
| Neutral | 23% (N=56) | 31% (N=90) |
| (Very) likely | 66% (N=159) | 56% (N=161) |
| *Increased scratching (z=−2.11, P=0.035)* | | |
| Not (very) likely | 5% (N=13) | 6% (N=18) |
| Neutral | 15% (N=35) | 22% (N=64) |
| (Very) likely | 80% (N=192) | 72% (N=208) |
| *Reduced play (z=−4.04, P<0.001)* | | |
| Not (very) likely | 1% (N=3) | 4% (N=13) |
| Neutral | 8% (N=19) | 18% (N=51) |
| (Very) likely | 91% (N=218) | 78% (N=226) |
| *Surface licking (z=−2.97, P=0.003)* | | |
| Not (very) likely | 20% (N=48) | 30% (N=86) |
| Neutral | 25% (N=61) | 27% (N=80) |
| (Very) likely | 55% (N=131) | 43% (N=124) |

pain signs (pain [very] likely: 53%, N=283 of 530 for dog owners versus 55%, N=64 of 117 for non-dog owners; P=0.618) and the third non-pain case (correctly identifying pain as not [very] likely in 91%, N=484 of 530 for dog owners versus 92%, N=108 of 117 for non-dog owners; P=0.061). For the second case of a dog with overt pain signs related to movement ability, dog owners rated pain to be a (very) likely motivation more so than non-dog owners (97%, N=513 of 530 for dog owners versus 92%, N=107 of 117 for non-dog owners; z=−2.57, P=0.010), see Table 4.

**Table 4. Case pain likeliness scores for a case of a dog displaying subtle pain signs, overt pain signs, and no pain signs, comparing dog owners and non-dog owners. Mann-Whitney U tests indicated differences or lack thereof for 530 dog owners and 117 non-dog owners (P<0.05).**

|  | Dog owners (N=530) | Non-dog owners (N=117) |
|---|---|---|
| *Case 1 – subtle pain signs (z=−0.50, P=0.618)* |  |  |
| Not (very) likely | 29% (N=151) | 25% (N=29) |
| Neutral | 18% (N=96) | 20% (N=24) |
| (Very) likely | 53% (N=283) | 55% (N=64) |
| *Case 2 – overt pain signs (z=−2.57, P=0.010)* |  |  |
| Not (very) likely | 2% (N=12) | 3% (N=4) |
| Neutral | 1% (N=5) | 5% (N=6) |
| (Very) likely | 97% (N=513) | 92% (N=107) |
| *Case 3 – no pain (z=−1.87, P=0.061)* |  |  |
| Not (very) likely | 79% (N=149) | 87% (N=102) |
| Neutral | 12% (N=65) | 5% (N=6) |
| (Very) likely | 9% (N=46) | 8% (N=9) |

### The three cases – participants' painful experiences or dogs' painful experiences

When assessing differences in possible pain recognition for the three cases between a) participants with a personal painful experience and those without and b) dog owners with experience with a dog's painful experience and those without, significant differences were found between the groups with and without the experiences for the first case of a dog with subtle pain signs, as following.

Regarding participants' own painful experiences, participants with a personal painful experience had higher frequencies of scoring pain as (very) likely than those without such experience, for the first case of a dog with subtle pain signs (60%, N=216 of 363 versus 46%, N=119 of 261; P<0.001; N=23 missing values). We found no significant differences for the second case of a dog with less subtle pain signs (97%, N=352 of 363 for participants with a previous painful experience versus 95%, N=248 of 261 for participants without such experience; P=0.2; N=23 missing values). Nor did we find significant differences for the third non-pain case (correctly identifying pain as not [very] likely in 80%, N=290 of 363, for participants with a previous painful experience versus 80%, N=210 of 261 for participants with a previous painful experience; P=0.8; N=23 missing values). See Table 5.

For dog-owning participants, those who had a dog experiencing a painful event previously, scored pain as (very) likely at significantly higher percentages for the first case of a dog with subtle pain signs, than those lacking such experience (62%, N=149 of 240 versus 46%, N=134 of 290; z=−3.53, P<0.001). For the second case these percentages were 98% (N=236 of 240) versus 96% (N=277 of 290; P=0.07). Interestingly, the participants with a previously painful dog also scored higher, but still low, percentages of indicating pain as (very) likely in the third, non-pain related case than those without (12%, N=28 of 240 versus 6%, N=18 of 290; z=−1.43, P=0.2). See Table 6.

### Discussion

The welfare of dogs in their role as companion animal is highly dependent on their caregivers. During their life span, companion dogs represent an important social partner to their owners and they may even take on the role of a family member [39,40]. Despite a possible close relationship, dog owners may find themselves challenged when assessing dog

**Table 5. Case pain likeliness scores for a case of a dog displaying subtle pain signs, overt pain signs, and no pain signs, comparing participants with and without a previous painful experience. Mann-Whitney U tests indicated differences or lack thereof for 363 participants with a previous painful experience and 261 without such experience (P<0.05).**

| | Previous painful experience (N=363) | No such previous experience (N=261) |
|---|---|---|
| *Case 1 – subtle pain signs (z=−3.75, P<0.001)* | | |
| Not (very) likely | 22% (N=81) | 35% (N=91) |
| Neutral | 18% (N=66) | 19% (N=51) |
| (Very) likely | 60% (N=216) | 46% (N=119) |
| *Case 2 – overt pain signs (z=−1.26, P=0.209)* | | |
| Not (very) likely | 2% (N=6) | 3% (N=8) |
| Neutral | 1% (N=5) | 2% (N=5) |
| (Very) likely | 97% (N=352) | 95% (N=248) |
| *Case 3 – no pain (z=−0.25, P=0.805)* | | |
| Not (very) likely | 80% (N=290) | 80% (N=210) |
| Neutral | 11% (N=40) | 12% (N=31) |
| (Very) likely | 9% (N=33) | 8% (N=20) |

**Table 6. Case pain likeliness scores for a case of a dog displaying subtle pain signs, overt pain signs, and no pain signs, comparing dog owners with and without a dog that previously suffered a painful experience. Mann-Whitney U tests indicated differences or lack thereof for 530 dog owners (P<0.05).**

| | Dog with previous painful experience (N=240) | Dog without such previous experience (N=290) |
|---|---|---|
| *Case 1 – subtle pain signs (z=−3.53, P<0.001)* | | |
| Not (very) likely | 23% (N=55) | 33% (N=96) |
| Neutral | 15% (N=36) | 21% (N=60) |
| (Very) likely | 62% (N=149) | 46% (N=134) |
| *Case 2 – overt pain signs (z=−1.83, P=0.068)* | | |
| Not (very) likely | 1% (N=3) | 3% (N=9) |
| Neutral | 0.4% (N=1) | 1% (N=4) |
| (Very) likely | 98% (N=236) | 96% (N=277) |
| *Case 3 – no pain (z=−1.43, P=0.151)* | | |
| Not (very) likely | 77% (N=184) | 81% (N=235) |
| Neutral | 12% (N=28) | 13% (N=37) |
| (Very) likely | 12% (N=28) | 6% (N=18) |

welfare [41]. Such challenges may include a difficulty to assess a dog's possible pain experience. Protection from pain is integrated in the animal welfare policies of several countries, for animals under human care [42]. To protect an animal from pain, correct pain assessment is a necessity. Pain assessment may be challenged by interspecies differences in pain expression [43] and the possible dog behavioural adaptation to the presence of humans [30,44]. Thus, studying how

humans recognise dog pain is relevant. We studied this recognition in dog owners and non-dog owners and assessed how they rated the likeliness of pain for seventeen behavioural signs and in three cases of changed dog behaviour. We found confirmation of our first hypothesis that dog owners do not recognise subtle dog pain signs more so than non-dog owners. However, we stress that our sample of dog owners and non-dog owners differed in demographic characteristics, such as age, as discussed below when we provide detail on the limitations of this study. Also, our non-dog owning sample consisted of people presently not owning a dog. Consequently, previous experience with dogs could have affected our non-dog owning sample. Taking into account these limitations of our study and the particularities of our sample, we found that dog owners achieved better results for identifying a dog case presenting overt dog pain signs relating to movement ability. Yet, both dog owners and non-dog owners rated pain likeliness high for this case, thus limiting potential benefits for dog welfare. A previous study also found movement-based pain signs to be recognisable to dog owners [45]. The difficulty in recognising the subtle pain signs that our results point at, could possibly stem from misinterpretation of more common dog behavioural signs. Previous research found dog pain signs easily misinterpreted as day-to-day behavioural changes [31]. Generally, people may struggle to interpret a dog's behaviour accurately and objectively, as supported by their focus on tail movements, which are not known to inform specifically on a dog's possible pain experience [22]. Additionally, a found tendency to provide more holistic interpretations of a dog's wants or feelings [22], could also explain a hampered interpretation of subtle pain signs.

Regarding specific behavioural signs, dog owners less so indicated a likeliness of pain for two, but not all, signs, that are also or alternatively known as stress/fear signs, such as lip/nose licking or yawning [46–48]. Behaviours of the dog's turning head or body away and freezing were indicated as less likely a sign of pain in dog owners than non-dog owners. No significant difference was found for hesitant paw lifting, lip/nose licking and yawning. Possibly, dog owners are generally more aware of certain stress/fear signs than of pain signs, thus assigning other causes than pain to possible subtle pain behaviours. Significantly higher correct identification of fearful dogs by more dog experienced people may further underpin this line of reasoning [49]. Alternatively, dog owners may find less common behavioural signs more likely a possible indication of pain, such as surface licking. In dogs the licking of surfaces is less seen than turning head/body away or freezing. (Excessive) surface licking, outside of, e.g., licking an empty food bowl, was indicated in a study on gastrointestinal disease to be a poorly documented pain sign, yet without a known prevalence [50,51]. It was classified as 'abnormal' indicating a lesser commonality [50,51].

We saw our second hypothesis partially confirmed. Participants with a personal painful experience recognised subtle dog pain signs more so than those without such experience, but only at significant levels for the behaviours of a dog's changed personality, increased blinking and yawning. Changed personality stood out for its relatively high levels of recognition. While reduced playfulness is a commonly recognised sign of pain experience in dogs [52], changes in personality and fluctuating mood are more typical human reactions to pain [53]. We did not assess for the seventeen possible pain signs, why people attributed pain as more or less likely, so we cannot determine reasons for the selection of particularly these behaviours being perceived as more likely dog pain related. It would be interesting to assess such reasons, as for instance to determine if empathy [54], or general pain attitude and knowledge [55,56], factor into dog pain sign recognition.

Our third hypothesis was underpinned by the results. Owners of dogs that suffered a previous painful event showed higher recognition of subtle pain signs and of pain as the cause of behavioural change, as compared to owners of dogs without such experience. This may indicate a general higher awareness of behavioural changes potentially being motivated by a dog's pain. Possibly even small educational efforts may have benefitted these dogs and their owners. For example, an educational intervention targeting several pain signs, using an adapted veterinary dog pain instrument, increased dog owners' ability to interpret dog behaviours possibly indicative of pain [57]. If even small educational intervention effort can increase the likeliness of timely recognition of a dog's pain signs, dogs and their owners can benefit of such intervention efforts.

Our study has several limitations. We asked participants only to rate the likeliness of pain, not of other forms of discomfort. As indicated in the Methods section, we deemed this necessary to define an easily understandable study purpose and limit questionnaire length. Asking participants to rate likeliness of pain *and* discomfort could have led to lower response rates, particularly from lesser topic-engaged participants as it would come with double the effort of partaking. Yet, we thus deviated from the approach taken by Mills et al. [29], who opted to use the term 'discomfort' to refer to pain and/or paraesthesia and/or dysesthesia. It is important to note that not every form of discomfort can be attributed to pain. Other, non-pain related, psychological and physical factors can potentially lead to the experience of discomfort [58]. However, combined measurement of pain and discomfort, is found in studies [59,60], referring to the description of pain being the main cause for discomfort [61]. By using only the term 'pain' in our questionnaire, we aimed to ensure that participants evaluated the examples given in terms of their possible association with pain, as defined by the International Association of the Study of Pain (IASP), describing pain as an 'unpleasant sensory and emotional experience, resembling or associated with, an actual or potential tissue damage' [62]. Future studies should look into the concept of discomfort, other than pain experience, and include a larger participant-base of non-dog owners. The larger participant-base of non-dog owners is needed as another limitation of our study is a skewed data set, that includes more dog owners than non-dog owners and is characterised by an uneven age and gender distribution.

Approximately half of our dog owning participants were aged between fifty and sixty-five years old, about half our non-dog owners were between eighteen and thirty-five years old and in our study 88% of participants identified as female. A more even age distribution, and a more evenly spread gender distribution are recommended for future research. Gender disparities in terms of a disproportionately high number of female participants are common in the study of human-animal interactions [63]. Previous research suggests females to be more sensitive to affective information than male observers [64] and thus uneven gender distribution can affect study outcomes. Our convenience-based study does not allow for assessment of the presence or absence of a (confounding or modifying) relationship between dog ownership, participants' own painful experiences and the ability to recognise behavioural pain signs in dogs. We suggest future studies to look into such possible confounding or modifying relationships as well as any possible relationships between human characteristics such as age, gender, animal husbandry experience and dog school attendance for dog owners and recognition of possible dog pain signs, particularly subtle pain signs as a next step towards insights that may ultimately benefit education on pain sign recognition.

We did not offer free text fields, as output from free text fields is varied and requires categorizing the output, making assumptions and risking categorization errors. As our aim was to study differences between sub samples, for specifically rating pain likeliness for seventeen previously indicated possible behavioural pain signs [29] and three cases describing less subtle, movement-related signs or subtle possible pain signs, we felt that categorization errors should be avoided to reach this aim. As a consequence, the study will likely have directed and structured thoughts of participants, losing the chance to freely capture thoughts on offered pain signs recognition opportunities, such as to arise when reading the three cases.

A final limitation of our study was the use of three dog behavioural cases that were not previously used in studies, but designed by us as vignettes for the specific purpose of our study. The cases were pretested and we opted for a Shepherd dog and a Chihuahua based on the causes of pain frequenting these breeds. We opted to not use similar dog breed types for the cases to prevent confusion or boredom in our participants. Yet, people are known to rate a dog's pain sensitivity differently depending on breed type [65] and this may have affected participants judgements of pain signs. We did not deem photographic support for the cases to be a strict necessity and thus did not include any photos, to prevent possible bias based on emotions elicited, or memories triggered by dog images. Yet, we recommend future studies to assess if case presentation affects possible judgement of behavioural signs and have made the case descriptions available in the supporting information of our article for this purpose.

Notwithstanding these limitations, our study points out that subtle pain signs in dogs may be difficult for people to recognise. Possibly, dog owners learned previously that these signs are underpinned by different motivations (also), hampering their identification as pain sign. This may affect dog health and welfare, if a dog is in pain and veterinary intervention is delayed to a point when pains signs are shown more overtly and therefore recognisably. As a consequence, it appears necessary to determine and potentially improve dog owners' abilities to recognise early signs of pain in their dogs, in order to prevent undetected suffering [66]. We suggest that dog owner education also addresses dog pain recognition, with a particular focus on subtle pain signs. In doing so, dog owners may be provided with early warning signs of pain and such early recognition may benefit dog welfare and prevent behavioural problems, including unwanted aggression.

## Supporting information

**S1 File. Three cases of dogs and their behavioural changes, presented to participants.**
(DOCX)

**S2 Table. The reported previous experience with a painful accident, illness and/ or treatment in N = 647 participants (N = 530 dog owners, N = 117 non-dog owners).**
(DOCX)

**S3 Table. The reported previous experience with their dog suffering a painful accident, illness and/ or treatment in N = 530 dog owners.**
(DOCX)

**S4 Table. The mean likeliness of a dog behavioural sign indicating pain as reported by N = 647 participants (N = 530 dog owners, N = 117 non-dog owners), with '0' indicating very unlikely and '4' indicating very likely.**
(DOCX)

**S5 Table. The likeliness percentages (N) of a dog behavioural sign indicating pain in three categories (not [very] likely, neutral, [very] likely scores) as reported by N = 647 participants and comparing dog owners (N = 530) to non-dog owners (N = 117) with Mann-Whitney U tests.**
(DOCX)

**S6 Table. The likeliness percentages (N) of a dog behavioural sign indicating pain in three categories (not [very] likely, neutral, [very] likely scores) as reported by N = 644 participants and comparing N = 363 participants indicating to have experienced a painful event themselves with N = 261 indicating not to have experienced this; with Mann-Whitney U tests.**
(DOCX)

**S7 Table. The likeliness percentages (N) of a dog behavioural sign indicating pain in three categories (not [very] likely, neutral, [very] likely scores) as reported by N = 530 dog owning participants and comparing N = 240 participants indicating to have a dog that previously experienced a painful event with N = 290 indicating their dog not to have experienced this; with Mann-Whitney U tests.**
(DOCX)

**S8 Table. The reported mean likeliness of a dog's motivation for the described behaviour in three cases and the reasons for a participant selecting the motivation with the highest likeliness in N = 647 participants (N = 530 dog owners, N = 117 non-dog owners; with '0' indicating very unlikely and '4' indicating very likely for the motivations and '0' indicating not a reason and '1' indicating a reason for the reasons for selecting a motivation at the highest likeliness).**
(DOCX)

**S9 Table. The reasons provided for attributing pain or learning processes as a motivation for described dog behaviour in three cases.**
(DOCX)

## Author contributions

**Conceptualization:** Silvia M. A. Gardeweg, Dionne E. Picard, Ineke R. van Herwijnen.

**Data curation:** Dionne E. Picard, Ineke R. van Herwijnen.

**Methodology:** Silvia M. A. Gardeweg, Dionne E. Picard, Ineke R. van Herwijnen.

**Software:** Dionne E. Picard.

**Writing – original draft:** Silvia M. A. Gardeweg, Dionne E. Picard, Ineke R. van Herwijnen.

**Writing – review & editing:** Ineke R. van Herwijnen.

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
