## [Decision Letter · Decision Letter 0]

24 Apr 2025

Dear Dr. Gardeweg,

We look forward to receiving your revised manuscript.

Kind regards,

I Anna S Olsson, Ph.D.

Academic Editor

PLOS ONE

Additional Editor Comments (if provided):

Reviewers' comments:

Reviewer's Responses to Questions

**Comments to the Author**

1. Is the manuscript technically sound, and do the data support the conclusions?

Reviewer #1: Yes

Reviewer #2: Partly

2. Has the statistical analysis been performed appropriately and rigorously?

Reviewer #1: I Don't Know

Reviewer #2: No

3. Have the authors made all data underlying the findings in their manuscript fully available?

Reviewer #1: Yes

Reviewer #2: Yes

4. Is the manuscript presented in an intelligible fashion and written in standard English?

Reviewer #1: No

Reviewer #2: Yes

Reviewer #1: The paper claims that dog owners struggle to interpret subtle signs of pain in dogs which could lead to welfare concerns for pet dogs by delaying diagnosis and treatment for painful conditions. The paper enhances the human-animal interaction and veterinary literature focusing on how people with and without experience owning a dog interpret dog behaviour. The paper focuses on pain behaviour which is challenging to recognise in non-human animals and is often understudied. The findings could be applied to educational interventions to increase pain recognition and ultimately improve the welfare of both dogs and their owners. The language is confusing in areas with some long convoluted sentences. The title and some subtitles are not concise and contain stacked nouns. The manuscript would benefit from more editing.

The introduction presents the lack of research in the area of pain behaviour recognition well but could have more critical evaluation of the current understanding of behaviours of pain and the positive and negative aspects of dog ownership. The hypotheses are clear but there was little explanation provided for why the authors hypothesised that dog owners would not recognise signs of pain in dogs more than non-dog owners. The authors may consider expanding on the hypothesis in a similar way as the hypothesis about pain experience.

The claims made are supported by the results and the data is presented in the supplementary material. The data for the observed frequencies of the combined Likert categories (likely/yes, unlikely/no) seem to be missing. Statistical analysis is lacking for the overall pain scores for the three case studies and the overall scores for the behaviours that participants attributed to pain.The authors carried out pairwise comparisons to identify where the differences lay between the two categories (likely, unlikely) between dog owners and non-dog owners. However, there is no mention of pairwise comparisons in the data analysis section. It would be useful to clarify whether the pairwise comparisons were carried out as a post-hoc analysis and consider carrying out a Bonferroni correction to adjust for type 1 error. The demographic information is thorough and presented well. More detail below:

33 - citation is missing for the statement on continuous growth of companion animal ownership

38 – The paragraph could be interpreted that pet ownership widely has a positive rather than negative impact on human health The sentence introducing the negative aspects of pet ownership makes it seem that there is consensus on the positive impact of pet ownership with few negative impacts (line 38 “next to positive effects, possible negative consequences of companion animal ownership may exist”). The impact of pet ownership on human health is debated in the field with recent articles claiming no evidence of benefits. I would suggest making it clear that there are mixed findings and adding mental health and stress as a negative aspect of dog ownership alongside mentioning the complex nature of pet ownership and its interplay with socio-demographic factors. One of the citations listed does not support the statement that companion animal ownership affects human health positively (line 37-38: Herzog et al 2011). The citation is better placed in the section that mentions the negative effects of companion animal ownership. I would also recommend incorporating the following paper: Mueller, M.K., King, E.K., Callina, K., Dowling-Guyer, S. and McCobb, E., 2021. Demographic and contextual factors as moderators of the relationship between pet ownership and health. Health Psychology and Behavioural Medicine, 9(1), pp.701-723.

44 – There is no citation to back claim that risk of dog bite increases when people fail to observe and interpret body language

45 – sentence confusing, needs rewording “many dog bites regard the context of people…”

48 – citation missing for claim that adequate recognition of warning signs decreases dog bite risks. This could also be re-worded to be less of an absolute claim such as ‘may require adequate…” as dog bites are complex and require multiple ways to mitigate.

64 – reword sentence, it is not clear what is being referred to with “this” and remove the personal opinion “worrisome”

75 – long sentence, break into multiple sentences for clarity

90 – “though” should be “through”

120 – there is no explanation provided for why the authors did not present the results. The part of the study presented in the paragraph is relevant to the entire study.

128 – remove “a” from “a pain sign”

137 – consider changing “raising” to “upbringing”

168 – clarify whether converting likert scores as mean scores refers to the matrix of 17 behaviours. This is descriptive analysis and requires some statistical analysis to assess if any differences

173 - There is no mention of the pair-wise comparison analysis carried out in the statistical analysis section

Analysis and results – Consider presenting the results of the chi-squared tests with the frequency (N=) alongside percentages. Consider presenting main findings as a graph for clarity.

Results – consider graphs to present data clearly as it can be difficult to visualise the data when presented as percentage in text. For example, age ranges and demographic data

224 – there is a lack of statistical analysis to compare the mean likelihood scores for scoring pain as a motivator between the three cases. The authors present descriptive analysis but no information about the significance of the difference between the groups.

219-245 – the paragraph presenting the results is difficult to understand and follow. A graph presenting the findings may help with clarity alongside rewording and shortening sentences.

294 – Present the pairwise comparison results or clarify whether the statistical result in the table refers to the comparison between unlikely or likely (maybe by placement next to the compared pair)

299 – “were” not “where”

322- in the text there is mention of a significant difference in recognition of pain sign “increased scratching”. In the table to p-value is 0.4 – this may be a typo

345 –It is not clear what “this” refer to in the beginning of the paragraph

380 – maybe mean “human” rather than “humane”?

418 – The limitations are listed well. There is a lack of detail about findings of previous studies on the impact of age on the interpretation of dog behaviour. Remove “thereof” at the end of the sentence.

422 – The reasoning provided for the skew towards female participants does not seem to explain why more women volunteer to participate than men. It may be worth mentioning that female participants are over-represented in human-animal interaction research (Herzog H (2021) Women dominate research on Human-Animal Bond. Am. Psychol ) and the proportion of female and male participants is similar to other studies (e.g. Hill, L., Winefield, H. and Bennett, P., 2020.; Westgarth et al 2018; Oxley et al 2018; Rohlf, V.I., Bennett, P.C., Toukhsati, S. and Coleman, G., 2010. Why do even committed dog owners fail to comply with some responsible ownership practices?. Anthrozoös, 23(2), pp.143-155.)

Reviewer #2: The overall aim of the paper is to assess areas of difficulty in the identification of dog pain signs and to identify differences between people with and without dog ownership experience. This area of research is useful and relevant to dog welfare. The hypotheses presented are 1 – that there is no difference in ability to identify subtle signs of dog pain between dog owners and non-owners; 2 – that there are differences in subtle dog pain sign recognition between those with and without personal pain experience; 3 – that there are differences in subtle dog pain sign recognition between those with and without experience owning a dog with a painful condition. The author’s research questions are interesting and could help target and refine future interventions for improving recognition of signs of pain in dogs.

Substantial revision is needed to improve flow and grammar, particularly in the introduction, and correct summarising of the literature needs to be checked, specifically in the section on perception of dog behaviour and emotional/motivational states. The methods section is clearly written and the introduction should be revised to achieve a similar quality. The methods section requires further detailing of definitions and classifications at several points (see detailed feedback below). While the authors have a well-collected data-set, the analyses presented do not appropriately account for population differences between the groups of interest (e.g., substantial differences in age distribution between the dog owner and non-dog owner populations). However, I believe this could be addressed with a reanalysis.

The authors have interesting data to present and have put substantial effort into the project, but I would suggest that major revisions would be needed prior to publication, both to the manuscript text and to the analysis.

Detailed feedback:

Abstract and Introduction:

- L11: Would recommend changing location of “in 530 dog owners and 117 non-dog owners” within the sentence to read smoother – perhaps “To investigate abilities and differences in dog pain sign recognition, we assessed these recognition skills through an online questionnaire in 530 dog owners and 117 non-dog owners.”

- L16: Significantly higher? Or provide more info regarding effect size as “substantial” is vague.

- L22: This sentence (“Dog owners less so than non-dog owners scored behaviours…” is a bit difficult to understand, particularly the last part regarding licking surfaces. Is it that dog owners were less likely to score ‘turning the head’… as possible pain signs, compared to non-dog owners, whereas they were more likely than non-dog owners to score ‘licking surfaces’ as a possible sign?

- L27: The first half of the last sentence of the abstract is difficult to understand upon first read through and would benefit from revision.

- L33: Is “according” the right word here?

- L45: “Many dog bites regard the context of people interacting with dogs…” This section of the sentence is not clear, a different word rather than “regard” may be needed?

- L53: While differences were found for tail movement, Wan et al reports the largest differences related to experience were reported for facial features (e.g., ears), not body features (pg 3 of Wan et al., 2012). See also Tami and Gallagher, 2009.

- L55: Amici et al., Törnqvist et al., and Bloom & Friedman are all specifically studying interpretation of facial expressions and are not a suitable citation for this argument. This is also contrary to Wan et al.’s findings that the biggest difference in which signals were used depending upon experience was in facial expressions, not general body signals.

- L51 – L62: Please revise the section on recognition of emotional and motivational states to more accurately reflect the literature and ensure citations are correct and reflect the statements made. See also Konok et al 2015, Lakestani et al., 2014, Mariti et al., 2012,

- L69/70: Your first citation of this bunch (Cloutier et al., 2005) does not directly provide evidence for the statement you have made in the sentence (“In dogs and other animals, pain signs are often subtle as to avoid further vulnerability”). It does provide evidence for signs generally being subtle, but not that the reason for this is to avoid further vulnerability, which reads as the primary point of your sentence. I have not reviewed the other citations listed, but please make sure that the sources you have cited directly support your statement. Also consider rewriting the sentence so that the focus is on signs being subtle and therefore difficult to identify.

- L76-77: Fear and appeasement were not found to be labelled significantly more accurately than pain, as per section 3.1 of Guo et al. (quote from Guo et al.: “In particular, we found a significant main effect of emotion category (F(8.76, 3906.41) = 421.68, p <.001, ηp2 =.49) with neutral attracting the highest categorization accuracy, followed by happiness and anger (p = 0.9), then anticipation and surprise (p = 1), then distress and sadness (p = 1), then fear, appeasement and pain (all ps > 0.05), and finally by frustration (neutral > happiness = anger > anticipation = surprise > distress = sadness > fear = appeasement = pain > frustration; post hoc multiple comparisons of emotion category between different groups, all ps < 0.01; Table 2).”)

- L90-92. Please revise as this is not a full sentence – I think “though” may be a typo of “through”?

Methods:

- L125 – Was a definition of pain provided to the participants?

- L128 – missing word “as a sign of pain” or “as a pain sign”

- L136 – The definition of the word “motivation” feels loosely defined here. “The dog’s raising” in particular is an unclear category and would include many instances of learned behaviour (i.e., “learning processes”). What definitions were provided to participants? Who were case studies written by? Were they real case studies or fabricated for the purpose of this study? How was a “ground truth” for the underlying “motivation” determined? How were they validated?

- L160: Repetition of “for the larger project” in this sentence

- L170: Why was “neutral” lumped with “very unlikely” and “unlikely” when it is the middle point between those options and “likely”/”very likely”?

- L170-174: Why not use a Mann – Whitney U test so you don’t need to collapse the “neutral” category in with the “Unlikely” category? By using a chi square test you’re losing granularity in the data and also treating the data as nominal rather than ordinal.

- Methods generally: The grammar and flow in this section are very good; please revise other sections of the paper in the same manner. More detail should be added regarding how participants were grouped based on ownership status (was it only current ownership, or also past?), dog pain experience, and personal pain experience. These classifications are currently only vaguely presented in the methods and it should be clarified what the definitions are and what definitions were presented to participants. The questions are also not presented in the appendix, which makes it difficult for readers to know what specifically was asked of participants and how these groups were created.

Results:

- L185 – Does this mean that all the other non-dog owning participants never walked dogs? May be worth clarifying as otherwise a reader could interpret the text to meanthat the other option was “more than weekly” rather than “never”.

- L189-191 Would suggest rephrasing as “but dogs owned by participants were more likely to be neutered…”

- L196-197: How did you account for this difference in age distributions between non-dog owners and owners in your analysis comparing scoring between dog-owners and non-dog owners? I.e., how do you know differences in the groups are due to ownership status and not age?

- L199 – L200: Please also provide the percentages of male/female participants for dog-owners so that readers can more easily compare between the two samples. As in previous comment, how was this difference in population accounted for between the dog-owner and non-dog owner groups?

- L210-L212: There is a difference in percentage of those having previous experience with a painful accident, illness, or treatment who are and are not dog owners of 12%. You hypothesize that both dog ownership and experience with a painful accident, illness, or treatment are correlated with ability to assess signs of pain in dogs. How do you disentangle these two effects?

- L223-225: The sentence does not accurately represent the data presented – the authors state that the motivation of pain has a lower likeliness score in the first case (2.4 +/- 1.3) than in the second case (3.7 +/- 0.7), which is accurate, but also than in the third “non-pain” case (0.7 +/- 1.0) which is incorrect based on the means they have presented.

- L218-L245: During study design, did you consider having participants provide free-text responses about which signs of pain they used to make their assessment rather than providing them with a list of signs that were present in the text? Participants may have an easier time identifying signs of pain from the text (or realising what signs they should have noticed) if they are provided alongside the question as has been done. This should be discussed/considered a bit during interpretation.

- L242-244: This is not a full sentence, please revise.

- L278 – 280: Can you clarify where these cut-offs (3.1 and 1.8) come from? You say three behaviours “scored higher mean pain”… and that three had “lower mean likeliness scores”. What is this in comparison to? If you are just presenting the three highest rated and three lowest rated behaviours I would suggest rewriting, possibly along the lines of “The three behaviours with the highest mean pain likeliness scores were ‘change in personality’ (mean +/- SD), ‘hesitant paw-lifting’ (mean +/- SD), and ‘fluctuating mood’ (mean +/- SD)…” with similar text for the three lowest rated.

- L315: Please revise this heading for clarity, specifically use of the word “depending”

- L330: It’s unclear which behaviours this sentence is referring to – is it all the behaviours? This would be incorrect though since increased scratching did not show a significant difference.

- Overall results: The order the study components are presented in the results section is not the same as in the methods section. The results may benefit from being reordered to align with the method section. There are also many pairwise comparisons being completed, but there is no adjustment made to the alpha level to account for multiple comparisons, meaning there is an increased risk for false positives.

Discussion:

- L339: This doesn’t quite read as a full sentence; please revise. (Sentence: “This, although protection from pain for animals under human care is integrated int eh animal welfare policies of several countries”.)

- L361: Methodological decisions regarding grouping “neither” with “unlikely” may be influencing results. Especially for signals which could be indicators of both stress/fear and pain, individuals may choose “neither” rather than selecting “unlikely” or “likely”. By grouping neither with “unlikely”, you may have artificially inflated the number of dog-owners selecting “unlikely” for signs which they responded to with “neither” due to being signs of both fear and pain. As you’ve suggested in the discussion, non-dog owners may be less aware of these signs also being potential indicators of fear.

o If your suggestion regarding this difference between dog owner and non-dog owners being due to the signals also being signs of fear/stress, would you have expected similar results for other signs of fear that are in the list of 17 behaviours you tested (e.g., lip licking, nose licking, hesitant paw-lifting)? Why or why not?

- L375: I would revise to avoid the use of the word “proof” as this is a very strong statement.

- L398: Was a clear definition of pain provided to the participants? Discussion of the definition used should be clearly detailed in L125 of the Methods where signposting to the discussion is present. A definition is provided in line 410. It would be good to clarify if this was the definition provided to participants, and to include this in the methods.

- L399: “As indicated in the methods section.” is not a complete sentence. Please revise the start of this paragraph.

- L415: It is very good that these limitations are highlighted here, but I think more can be done in the analysis itself to account for the differing age distributions and other issues I’ve highlighted in comments on the methods.

- L429: Thank you for mentioning the differences in perception of pain by breed, this is important to include here.

- Bibliography: Please check formatting, as the citation format of journal titles is not standardised – capital case in journal titles, abbreviations used in journal titles, etc.

**Do you want your identity to be public for this peer review?** For information about this choice, including consent withdrawal, please see our Privacy Policy

Reviewer #1: No

Reviewer #2: No

---

## [Author Response · Author response to Decision Letter 1]

18 Jun 2025

Dear Reviewers,

Thank you for your valuable feedback which we used to improve our manuscript accordingly. You may find details on the improvements below. Please note that lines are indicated for the manuscript without track changes, to facilitate ease of reading.

Reviewer #1:

The data for the observed frequencies of the combined Likert categories (likely/yes, unlikely/no) seem to be missing.

Thank you for this suggested addition to the already presented means ± S.D. We have added these as S5 Appendix and refer to the appendix in the text (line 250).

Statistical analysis is lacking for the overall pain scores for the three case studies and the overall scores for the behaviours that participants attributed to pain.

Thank you, we have rewritten the section on the three cases in light of your feedback and other reviewer feedback and hope to provide a more complete section now (starting at line 326).

The authors carried out pairwise comparisons to identify where the differences lay between the two categories (likely, unlikely) between dog owners and non-dog owners. However, there is no mention of pairwise comparisons in the data analysis section. It would be useful to clarify whether the pairwise comparisons were carried out as a post-hoc analysis and consider carrying out a Bonferroni correction to adjust for type 1 error. The demographic information is thorough and presented well.

Thank you for the complement and Bonferroni correction was applied. The methods section now more accurately reflects the information on the Chi-square testing. (Line 187 onwards: ‘For these combined scores, we tested for differences a) between the dog owners and non-dog owners with Pearson’s Chi-square tests and b) between previous painful experiences or lack thereof in 1) participants and 2) dogs of dog-owning participants, regarding Bonferroni method adjusted P-values of <0.05 as statistically significant.’)

33 - citation is missing for the statement on continuous growth of companion animal ownership

Thank you, citations are now behind both the sentence starting in line 34 and the sentence starting at line 36.

38 – The paragraph could be interpreted that pet ownership widely has a positive rather than negative impact on human health The sentence introducing the negative aspects of pet ownership makes it seem that there is consensus on the positive impact of pet ownership with few negative impacts (line 38 “next to positive effects, possible negative consequences of companion animal ownership may exist”). The impact of pet ownership on human health is debated in the field with recent articles claiming no evidence of benefits. I would suggest making it clear that there are mixed findings and adding mental health and stress as a negative aspect of dog ownership alongside mentioning the complex nature of pet ownership and its interplay with socio-demographic factors. One of the citations listed does not support the statement that companion animal ownership affects human health positively (line 37-38: Herzog et al 2011). The citation is better placed in the section that mentions the negative effects of companion animal ownership. I would also recommend incorporating the following paper: Mueller, M.K., King, E.K., Callina, K., Dowling-Guyer, S. and McCobb, E., 2021. Demographic and contextual factors as moderators of the relationship between pet ownership and health. Health Psychology and Behavioural Medicine, 9(1), pp.701-723.

Thank you for your comment and suggestions. We have revised the respective section, starting at line 37, and added the suggested reference.

44 – There is no citation to back claim that risk of dog bite increases when people fail to observe and interpret body language.

Thank you, we have added two citations (middle of line 49).

45 – sentence confusing, needs rewording “many dog bites regard the context of people…”

We have adapted this sentence to improve its understanding, starting in line 48.

48 – citation missing for claim that adequate recognition of warning signs decreases dog bite risks. This could also be re-worded to be less of an absolute claim such as ‘may require adequate…” as dog bites are complex and require multiple ways to mitigate.

Thank you, we have reworded the sentence according to your suggestion: “If interactions can be stopped timely or avoided altogether in risk contexts, biting risk might decrease, but this may require adequate recognition of warning signs in a dog’s body language.” (lines 51-54).

64 – reword sentence, it is not clear what is being referred to with “this” and remove the personal opinion “worrisome”

Thank you, this text section was adapted, also to adhere to the second reviewers comments regarding the Introduction section.

75 – long sentence, break into multiple sentences for clarity

We have split this sentence into two sentences.

90 – “though” should be “through”

Thank you, we have corrected this spelling error.

120 – there is no explanation provided for why the authors did not present the results. The part of the study presented in the paragraph is relevant to the entire study.

Thank you for sharing your interest in the images and pointing to the opportunity to explain why we do not incorporate these outcomes in this article. We have added this explanation in line onwards: ‘In the present study we did not use the study results from these images, as we aim to study these further in a separate study and focused the present article on pain sign recognition based on the descriptions of pain signs and cases.’

137 – consider changing “raising” to “upbringing”

Thank you, we have changed this term.

168 – clarify whether converting likert scores as mean scores refers to the matrix of 17 behaviours. This is descriptive analysis and requires some statistical analysis to assess if any differences

Thank you, we have now more clearly mentioned that this refers to the behavioural signs in line 184 onwards.

173 - There is no mention of the pair-wise comparison analysis carried out in the statistical analysis section

Thank you for addressing that the methods section could be improved - it now more accurately reflects the information on the Chi-square testing, as also indicated earlier on in this response to reviewer letter.

Analysis and results – Consider presenting the results of the chi-squared tests with the frequency (N=) alongside percentages. Consider presenting main findings as a graph for clarity.

Thank you for this suggestion, we now provide the frequencies throughout the text. We had tested figures with several people pre-reading the article and they did not feel these added value, making us opt to not work with figures but tables.

224 – there is a lack of statistical analysis to compare the mean likelihood scores for scoring pain as a motivator between the three cases. The authors present descriptive analysis but no information about the significance of the difference between the groups.

Thank you for your suggestion. We had opted to add this as a supplement. But have now added statistical information, rewriting the paragraph.

219-245 – the paragraph presenting the results is difficult to understand and follow. A graph presenting the findings may help with clarity alongside rewording and shortening sentences.

Thank you for your suggestion, we have carefully reconsidered our previous input from pre readers and rediscussed with two of them and based on this feedback, we would like to keep the information in texts and tables only.

294 – Present the pairwise comparison results or clarify whether the statistical result in the table refers to the comparison between unlikely or likely (maybe by placement next to the compared pair)

Thank you for addressing that the methods section could be improved - it now more accurately reflects the information on the Chi-square testing.

299 – “were” not “where”

Thank you, we have corrected this spelling error.

322- in the text there is mention of a significant difference in recognition of pain sign “increased scratching”. In the table to p-value is 0.4 – this may be a typo

Thank you for pointing out the error in the table. We checked the table and fixed the error.

345 –It is not clear what “this” refer to in the beginning of the paragraph

Thank you for addressing the unclear reference to earlier text, which we have solved by rewriting the start of the discussion section.

380 – maybe mean “human” rather than “humane”?

We have changed this wording accordingly.

418 – The limitations are listed well. There is a lack of detail about findings of previous studies on the impact of age on the interpretation of dog behaviour. Remove “thereof” at the end of the sentence.

We have removed the word ‘thereof’.

422 – The reasoning provided for the skew towards female participants does not seem to explain why more women volunteer to participate than men. It may be worth mentioning that female participants are over-represented in human-animal interaction research (Herzog H (2021) Women dominate research on Human-Animal Bond. Am. Psychol ) and the proportion of female and male participants is similar to other studies (e.g. Hill, L., Winefield, H. and Bennett, P., 2020.; Westgarth et al 2018; Oxley et al 2018; Rohlf, V.I., Bennett, P.C., Toukhsati, S. and Coleman, G., 2010. Why do even committed dog owners fail to comply with some responsible ownership practices?. Anthrozoös, 23(2), pp.143-155.)

Thank you, we have added this information, based on your suggested main source, in lines 453-455.

Reviewer #2:

Abstract and Introduction:

- L11: Would recommend changing location of “in 530 dog owners and 117 non-dog owners” within the sentence to read smoother – perhaps “To investigate abilities and differences in dog pain sign recognition, we assessed these recognition skills through an online questionnaire in 530 dog owners and 117 non-dog owners.”

Thank you, we have exchanged this sentence with your suggested formulation.

- L16: Significantly higher? Or provide more info regarding effect size as “substantial” is vague.

Thank you for this suggestion, the word was replaced.

- L22: This sentence (“Dog owners less so than non-dog owners scored behaviours…” is a bit difficult to understand, particularly the last part regarding licking surfaces. Is it that dog owners were less likely to score ‘turning the head’… as possible pain signs, compared to non-dog owners, whereas they were more likely than non-dog owners to score ‘licking surfaces’ as a possible sign?

Thank you for your comment, the sentence in the abstract (starting at line 17 of the manuscript without track changes) has been reworded to: “The behaviours of ‘turning the head or body away’ and ‘freezing’ were more often interpreted as possible pain signs by non-dog owners, whereas ‘licking surfaces’ was more often perceived as pain sign by dog owners.”

- L27: The first half of the last sentence of the abstract is difficult to understand upon first read through and would benefit from revision.

Thank you, we have revised the first half of the last sentence of the abstract: “Our finding that owners of dogs that experienced a painful event, scored a higher likeliness of certain possible pain signs, indicates that experience matters and this underpins how education on dog behaviour could possibly benefit animal welfare through addressing subtle pain signs in dogs.” (line 28 onwards)

- L33: Is “according” the right word here?

We have changed the wording to: “In view of the continuing growth” (line 34).

- L45: “Many dog bites regard the context of people interacting with dogs…” This section of the sentence is not clear, a different word rather than “regard” may be needed?

Thank you for addressing the unclarity. We have changed the sentence, starting at line 49 to: “Body language of a dog seems relevant, as often dog bites occur when people interact with (familiar) dogs, as was seen in familiar dogs causing a biting incident in 66% (N=320 of 484) of self-identified UK dog bite victim cases [21].”.

- L53: While differences were found for tail movement, Wan et al reports the largest differences related to experience were reported for facial features (e.g., ears), not body features (pg 3 of Wan et al., 2012). See also Tami and Gallagher, 2009. & - L55: Amici et al., Törnqvist et al., and Bloom & Friedman are all specifically studying interpretation of facial expressions and are not a suitable citation for this argument. This is also contrary to Wan et al.’s findings that the biggest difference in which signals were used depending upon experience was in facial expressions, not general body signals. & - L51 – L62: Please revise the section on recognition of emotional and motivational states to more accurately reflect the literature and ensure citations are correct and reflect the statements made. See also Konok et al 2015, Lakestani et al., 2014, Mariti et al., 2012.

Thank you for your feedback, the Introduction section was rewritten, also in line with the general feedback on improving the readability of the introduction.

- L69/70: Your first citation of this bunch (Cloutier et al., 2005) does not directly provide evidence for the statement you have made in the sentence (“In dogs and other animals, pain signs are often subtle as to avoid further vulnerability”). It does provide evidence for signs generally being subtle, but not that the reason for this is to avoid further vulnerability, which reads as the primary point of your sentence. I have not reviewed the other citations listed, but please make sure that the sources you have cited directly support your statement. Also consider rewriting the sentence so that the focus is on signs being subtle and therefore difficult to identify.

Thank you for this advice. We have rewritten this text, starting in line 56 to keep the focus on the signs being subtle and provided examples from a review to present the reader with examples.

- L76-77: Fear and appeasement were not found to be labelled significantly more accurately than pain, as per section 3.1 of Guo et al. (quote from Guo et al.: “In particular, we found a significant main effect of emotion category (F(8.76, 3906.41) = 421.68, p <.001, ηp2 =.49) with neutral attracting the highest categorization accuracy, followed by happiness and anger (p = 0.9), then anticipation and surprise (p = 1), then distress and sadness (p = 1), then fear, appeasement and pain (all ps > 0.05), and finally by frustration (neutral > happiness = anger > anticipation = surprise > distress = sadness > fear = appeasement = pain > frustration; post hoc multiple comparisons of emotion category between different groups, all ps < 0.01; Table 2).”)

Thank you for addressing this. We adapted the section also to adhere to the general feedback of improving the introduction section.

- L90-92. Please revise as this is not a full sentence – I think “though” may be a typo of “through”?

This issue was solved.

methods:

- L125 – Was a definition of pain provided to the participants?

Thank you for this question. A definition of pain was not provided, as not to influence the participants' initial perception and interpretation with a predetermined definition. We have now made this explicit in the methods section (lines 132-133).

- L128 – missing word “as a sign of pain” or “as a pain sign”

Thank you, this adaptation was made.

- L136 – The definition of the word “motivation” feels loosely defined here. “The dog’s raising” in particular is an unclear category and would include many instances of learned behaviour (i.e., “learning processes”). What definitions were provided to participants? Who were case studies written by? Were they real case studies or fabricated for the purpose of this study? How was a “ground truth” for the underlying “motivation” determined? How were they validated?

Our study aim was to assess if and how people may recognize behaviour signs in dogs as indicative of pain. We opted for the seventeen signs from Mills et al. (2023) and newly created cases in vignette format as an addition. As no already validated cases are available these vignettes were created based on descriptions of ailments in literature and established and checked by a dog behavioural spec

---

## [Decision Letter · Decision Letter 1]

3 Sep 2025

Dear Dr. Herwijnen,

Thank you for submitting your manuscript to PLOS ONE. After careful consideration, we feel that it has merit but does not fully meet PLOS ONE’s publication criteria as it currently stands. Therefore, we invite you to submit a revised version of the manuscript that addresses the minor points raised during the review process.

We look forward to receiving your revised manuscript.

Kind regards,

I Anna S Olsson, Ph.D.

Academic Editor

PLOS ONE

Journal Requirements:

Additional Editor Comments:

Reviewer #1:

Reviewers' comments:

Reviewer's Responses to Questions

**Comments to the Author**

Reviewer #1: All comments have been addressed

2. Is the manuscript technically sound, and do the data support the conclusions?

Reviewer #1: Yes

3. Has the statistical analysis been performed appropriately and rigorously?

Reviewer #1: Yes

4. Have the authors made all data underlying the findings in their manuscript fully available?

Reviewer #1: Yes

5. Is the manuscript presented in an intelligible fashion and written in standard English?

Reviewer #1: No

Reviewer #1: Thank you for responding to comments and making the suggested changes. The manuscript is much clearer but I think it would benefit from further editing as there are some sentences that are convoluted and challenging to understand. I have made a few specific suggestions below regarding the grammar but a more in depth edit would be required.

Please remove "as such" from the end of sentences (e.g line 277, 493)

line 153 - I would change "gender" to "sex" as I presume you are referring to the dog's sex and not the participant's gender.

line 210 - "were" not "was"

line 397- Good point. Could you add a citation for lip-liking and yawning as signs of stress/fear?

281 - Reads as if the person had a previous painful experience but I understand that the section is about people who had a dog with a painful condition. Maybe reword for clarity to avoid confusion.

Line 390- Remove "at least in part"

Line 403 - Remove "or the like"

**Do you want your identity to be public for this peer review?** For information about this choice, including consent withdrawal, please see our Privacy Policy

Reviewer #1: No

---

## [Author Response · Author response to Decision Letter 2]

6 Sep 2025

Utrecht, September 6th, 2025

Dear Mrs. Olsson,

Thank you again to yourself and the reviewers, for assessing the manuscript ‘The abilities in dog pain sign recognition as assessed by presenting seventeen listed dog behavioural signs and three case descriptions to dog owners and non-dog owners’ for publication in PLOS ONE.

We have made the specific adaptations requested by Reviewer #1 and have further edited sentences that could be convoluted and challenging to understand.

We look forward to the publication.

Dr. Silvia Gardeweg, Dionne Picard and Ineke van Herwijnen

---

## [Decision Letter · Decision Letter 2]

30 Oct 2025

Dear Dr. Herwijnen,

We look forward to receiving your revised manuscript.

Kind regards,

Carlos Alberto Antunes Viegas, DVM; MSc; PhD

Academic Editor

PLOS ONE

Journal Requirements:

Reviewers' comments:

Reviewer's Responses to Questions

**Comments to the Author**

Reviewer #1: All comments have been addressed

Reviewer #2: (No Response)

2. Is the manuscript technically sound, and do the data support the conclusions?

Reviewer #1: Yes

Reviewer #2: Partly

3. Has the statistical analysis been performed appropriately and rigorously?

Reviewer #1: Yes

Reviewer #2: No

4. Have the authors made all data underlying the findings in their manuscript fully available?

Reviewer #1: Yes

Reviewer #2: Yes

5. Is the manuscript presented in an intelligible fashion and written in standard English?

Reviewer #1: Yes

Reviewer #2: Yes

Reviewer #1: The study is very interesting and highlights the importance of studying differences of experiences with dogs and pain when assessing knowledge of dog behaviour. The results are presented very clearly and there is a sound discussion of limitations and future implications of the findings. I would suggest a few minor changes to improve readability in some areas and clarify some points which could be mis-understood. Please see my comments below:

51 – sentence starting “If interactions in risk contexts…” needs to be re-written as it is difficult to understand. Possibly Adequate recognition of warning signs from dog body language may prevent or stop risky interactions from occurring.

56 – I would change “science” has provided insights to "studies" or "research" or “in recent years there have been new insights”…

67 – re-word the first sentence of the paragraph to clarify. Example, The lack of studies exploring the ability of dog an non-dog owners in interpreting …

70 – It is great that there is a mention that dog bites may be caused by pain. Could you expand and add a citation? E.g. When in pain dogs may behave unpredictably and react to stimuli that they usually would not react to. Mills, D.S., Demontigny-Bédard, I., Gruen, M., Klinck, M.P., McPeake, K.J., Barcelos, A.M., Hewison, L., Van Haevermaet, H., Denenberg, S., Hauser, H. and Koch, C., 2020. Pain and problem behavior in cats and dogs. Animals, 10(2), p.318.

75 – It is unclear what “more subtle” is referring to. More subtle pain sign than what? Maybe remove more?

161: the sentences describing how participants were recruited is very informative, but I think it could be written in a clearer way. The word “indicating” could be replaced with “…describing that the study was about dog behaviour without specifying the study aim to assess pain recognition”.

164 – For readability maybe use partake only once in the sentence.

198- "that is" change to "aged between"

365 – comma not needed

398 - comma not needed

400 – reword for clarity – e.g. “ in dogs the licking of surfaces is less…than turning of the head/body or freezing”

402 – remove “a dog’s”

408 - remove “a dog’s”

409 – recognised more than what? Maybe write as being most recognised by….?

496 – the final sentence of the discussion is a bit difficult to follow as it is long. Is the main point to suggest dog owners receive education focusing not only on fear/stress recognition but also on recognising subtle signs of pain? Maybe then reiterate in a separate sentence the last point that this education may result in potentially benefitting….

Reviewer #2: Thank you for the extensive revisions. The manuscript is much improved, but I would like to follow up on several key comments which I would like to further discuss.

Comments and responses:

- L170: Why was “neutral” lumped with “very unlikely” and “unlikely” when it is the middle point between those options and “likely”/”very likely”?

Author response: Thank you for indicating this omission in the methods section. We have explained this in the methods now, line 184 onwards.

Reviewer response: While I understand that the author argues that neutral would count as “not indicating pain”, that may not be how a participant interpreted this category, since it is “neutral” and not one of the “unlikely” categories. A participant may have selected “neutral” where a behaviour could be indicative of pain in certain circumstances, but of fear (for example) in other circumstances and therefore chose the “Neutral” category. Given that the authors do not collapse the ordinal scale into a binary variable in other parts of the paper (means and standard deviations are reported), it seems odd that they would choose to do so here instead of using the full information they have available from their ordinal scale.

- L170-174: Why not use a Mann – Whitney U test so you don’t need to collapse the “neutral” category in with the “Unlikely” category? By using a chi square test you’re losing granularity in the data and also treating the data as nominal rather than ordinal.

Author response: We opted for the binary testing as this is in line with our study objective and we have many elements of testing, risking a too detailed text. To benefit reading and understanding, in line with our hypotheses and research question, this choice was made. We have now more clearly indicated this in the methods section line 184 onwards.

Reviewer response: Could the author please clarify what is meant by “we have many elements of testing, risking a too detailed text?” It’s not clear how a Mann-Whitney U test is harder to explain than a chi-square test and given that it’s appropriate for ordinal data, it would remove the need for explaining how/why ordinal data was collapsed into a binary variable. This would likely make the text shorter. A Mann-Whitney U test would tell you if there is a difference in how likely a member from group A is to consider a behaviour to be a sign of pain, compared to someone from group B. It’s not clear to me how this does not correctly align with the study objective? What you’ve said here also goes against the way you use mean and SD in other parts of the manuscript (e.g., lines 232-234 “Using the list of seventeen possible pain signs as explained in the Methods section, we assessed to which extent these were seen likely as indicative of pain by calculating means +/- S.D.”)

- L196-197: How did you account for this difference in age distributions between non-dog owners and owners in your analysis comparing scoring between dog-owners and non-dog owners? I.e., how do you know differences in the groups are due to ownership status and not age?

Author response: Thank you for pointing this out. As our study’s aim was not to indicate owner age differences in pain recognition we feel that adding another section to the already complex results incorporating a section where age splits are made and tested, will not be in line with study aim and further complicate our results section. This is why we opted to clearly point out in the discussion this skewness. In line with your valid comment, we have added to the discussion section (line 464 onwards) the suggestion for future research that looks into age and other possible participant characteristics that could factor in, such as dog school attendance as to ensure this limitation of our study is given sufficient attention by the readers of the article (and hoping the suggestion will be picked up in future studies of course!)

Reviewer response: I understand that there are already a lot of comparisons being run and splitting by age would be adding to this number. However, in lines 374 – 406 the results of the dog owner versus non-dog owner analyses are discussed, but the issue of the substantial differences in age distribution are not addressed at all. This is potentially misleading for a reader who is only skimming the paper and has skipped to the discussion, as this difference in demographics between dog owners and non-dog owners is a major limitation. The purpose of my comment was not that we should necessarily look into age effects (as this is not the variable of interest here), but that it is an issue that the results are discussed in a way that doesn't highlight the other major difference between the two groups. In this large section on the first hypothesis, the difference in age distributions between the two groups should be clearly mentioned and it should be pointed out that as this analysis cannot determine causal effect and that, difference between the group of dog owners versus non dog owners could be related other factors such as age. While this is mentioned briefly in the limitations (lines 449-454) I would prefer to see this discussed as part of the discussion section on that specific hypothesis, as this is a specific limitation of that analysis and significantly impacts interpretability. If someone wanted to address this in future, it would need to be done using something like a causal inference approach.

Additional reviewer comments:

The author should clarify how they accounted for multiple comparisons. They mention a correction for multiple comparisons in the text, but it is not clear if it is applied correctly. Using the data they provided, I ran a chi-square using the author's methods (grouping neutral with unlikely) for the dog owner vs non-dog owner comparison. The proportion of owners/non-owners identifying the signs as potentially indicative of pain are the same as what they report, but the chi-square statistics and p-values I’m getting are different with no correction applied. This would indicate to me that a correction was applied, but it’s particularly confusing that they report a p-value for both freezing and turning head/body away that is larger than what I’ve found (this could be because they accounted for multiple comparisons), but the p-value they report for licking surfaces is smaller than the p-value I calculated. So, if they’re correcting for multiple comparisons, it’s not clear how this is done, since whatever they’re adjusting their p-values by should be the same for all rows of the table (i.e., all behaviours assessed)?

P-values I calculated (with no correction for multiple comparisons for the three behaviours in question):

PS_freez: p = 0.00417

PS_turnaw: p = 0.00452

PS_licksurf = 0.04992

P-values reported in the paper (according to methods section, corrected for multiple comparisons):

Freezing: p = 0.03

Turning head or body away: p = 0.03

Licking surfaces: p = 0.04

The authors reported in lines 187-189 that Bonferroni method adjusted p-values were used. However, from my understanding in that case the p-values should all have been adjusted using the same multiplier, which it appears they have not been. Could the authors please clarify how this correction was applied?

**Do you want your identity to be public for this peer review?** For information about this choice, including consent withdrawal, please see our Privacy Policy

Reviewer #1: No

Reviewer #2: No

---

## [Author Response · Author response to Decision Letter 3]

18 Nov 2025

Utrecht, November 18th, 2025

Dear Academic Editor and Reviewers,

Thank you for your response that allows us to improve our manuscript titled ‘The abilities in dog pain sign recognition as assessed by presenting seventeen listed dog behavioural signs and three case descriptions to dog owners and non-dog owners’. We have carefully considered the provided feedback. Please find here below, our response to each point raised. Please note that line indications refer to the manuscript without track changes, for your easy reading.

Sincerely,

Ineke R. van Herwijnen Ph.D.

i.r.vanherwijnen@uu.nl

Division of Animals in Science and Society, Faculty of Veterinary Medicine, Department of Population Health Sciences, Utrecht University, Utrecht, The Netherlands

Response to each point raised:

Reviewer #1: The study is very interesting and highlights the importance of studying differences of experiences with dogs and pain when assessing knowledge of dog behaviour. The results are presented very clearly and there is a sound discussion of limitations and future implications of the findings. I would suggest a few minor changes to improve readability in some areas and clarify some points which could be mis-understood. Please see my comments below:

Thank you for your time, the kind feedback and suggestions for improvements.

51 – sentence starting “If interactions in risk contexts…” needs to be re-written as it is difficult to understand. Possibly Adequate recognition of warning signs from dog body language may prevent or stop risky interactions from occurring.

Thank you for this suggestion which we used to replace the sentence.

56 – I would change “science” has provided insights to "studies" or "research" or “in recent years there have been new insights”…

Thank you, we have replaced ‘science has’ for ‘studies have’

67 – re-word the first sentence of the paragraph to clarify. Example, The lack of studies exploring the ability of dog an non-dog owners in interpreting …

Thank you for this suggestion which we used to replace the first part of the sentence.

70 – It is great that there is a mention that dog bites may be caused by pain. Could you expand and add a citation? E.g. When in pain dogs may behave unpredictably and react to stimuli that they usually would not react to. Mills, D.S., Demontigny-Bédard, I., Gruen, M., Klinck, M.P., McPeake, K.J., Barcelos, A.M., Hewison, L., Van Haevermaet, H., Denenberg, S., Hauser, H. and Koch, C., 2020. Pain and problem behavior in cats and dogs. Animals, 10(2), p.318.

Thank you for the suggestion, which has been added (with the reference).

75 – It is unclear what “more subtle” is referring to. More subtle pain sign than what? Maybe remove more?

Thank you, we have removed ‘more’ in ‘more subtle’ throughout.

161: the sentences describing how participants were recruited is very informative, but I think it could be written in a clearer way. The word “indicating” could be replaced with “…describing that the study was about dog behaviour without specifying the study aim to assess pain recognition”.

Thank you for this suggestion which we used to replace the last part of the sentence.

164 – For readability maybe use partake only once in the sentence.

Indeed and we replaced the second ‘to partake’ by ‘to do so’.

198- "that is" change to "aged between"

Thank you, ‘that is’ was removed.

365 – comma not needed

Thank you, comma removed.

398 - comma not needed

Thank you, comma removed.

400 – reword for clarity – e.g. “ in dogs the licking of surfaces is less…than turning of the head/body or freezing”

Thank you for this suggestion which we used to replace the first part of the sentence.

402 – remove “a dog’s” & 408 - remove “a dog’s”

Removed.

409 – recognised more than what? Maybe write as being most recognised by….?

Thank you for pointing this out, the sentence was adapted to ‘stood out for their relatively high levels of recognition’.

496 – the final sentence of the discussion is a bit difficult to follow as it is long. Is the main point to suggest dog owners receive education focusing not only on fear/stress recognition but also on recognising subtle signs of pain? Maybe then reiterate in a separate sentence the last point that this education may result in potentially benefitting….

Thank you for addressing this. The last sentences now read: ‘We suggest that dog owner education also addresses dog pain recognition, with a particular focus on subtle pain signs. In doing so, dog owners may be provided with early warning signs of pain. Early recognition of pain in dogs will benefit dog welfare and may also prevent behavioural problems such as aggression.’

Reviewer #2:

Reviewer #2: Thank you for the extensive revisions. The manuscript is much improved, but I would like to follow up on several key comments which I would like to further discuss.

Thank you for your time and extensive feedback. We have addressed this as following.

- L170: Why was “neutral” lumped with “very unlikely” and “unlikely” when it is the middle point between those options and “likely”/”very likely”?

Author response: Thank you for indicating this omission in the methods section. We have explained this in the methods now, line 184 onwards.

Reviewer response: While I understand that the author argues that neutral would count as “not indicating pain”, that may not be how a participant interpreted this category, since it is “neutral” and not one of the “unlikely” categories. A participant may have selected “neutral” where a behaviour could be indicative of pain in certain circumstances, but of fear (for example) in other circumstances and therefore chose the “Neutral” category. Given that the authors do not collapse the ordinal scale into a binary variable in other parts of the paper (means and standard deviations are reported), it seems odd that they would choose to do so here instead of using the full information they have available from their ordinal scale.

- L170-174: Why not use a Mann – Whitney U test so you don’t need to collapse the “neutral” category in with the “Unlikely” category? By using a chi square test you’re losing granularity in the data and also treating the data as nominal rather than ordinal.

Author response: We opted for the binary testing as this is in line with our study objective and we have many elements of testing, risking a too detailed text. To benefit reading and understanding, in line with our hypotheses and research question, this choice was made. We have now more clearly indicated this in the methods section line 184 onwards.

Reviewer response: Could the author please clarify what is meant by “we have many elements of testing, risking a too detailed text?” It’s not clear how a Mann-Whitney U test is harder to explain than a chi-square test and given that it’s appropriate for ordinal data, it would remove the need for explaining how/why ordinal data was collapsed into a binary variable. This would likely make the text shorter. A Mann-Whitney U test would tell you if there is a difference in how likely a member from group A is to consider a behaviour to be a sign of pain, compared to someone from group B. It’s not clear to me how this does not correctly align with the study objective? What you’ve said here also goes against the way you use mean and SD in other parts of the manuscript (e.g., lines 232-234 “Using the list of seventeen possible pain signs as explained in the Methods section, we assessed to which extent these were seen likely as indicative of pain by calculating means +/- S.D.”)

We now present data split in three categories and using Mann-Whitney U tests.

- L196-197: How did you account for this difference in age distributions between non-dog owners and owners in your analysis comparing scoring between dog-owners and non-dog owners? I.e., how do you know differences in the groups are due to ownership status and not age?

Author response: Thank you for pointing this out. As our study’s aim was not to indicate owner age differences in pain recognition, we feel that adding another section to the already complex results incorporating a section where age splits are made and tested, will not be in line with study aim and further complicate our results section. This is why we opted to clearly point out in the discussion this skewness. In line with your valid comment, we have added to the discussion section (line 464 onwards) the suggestion for future research that looks into age and other possible participant characteristics that could factor in, such as dog school attendance as to ensure this limitation of our study is given sufficient attention by the readers of the article (and hoping the suggestion will be picked up in future studies of course!)

Reviewer response: I understand that there are already a lot of comparisons being run and splitting by age would be adding to this number. However, in lines 374 – 406 the results of the dog owner versus non-dog owner analyses are discussed, but the issue of the substantial differences in age distribution are not addressed at all. This is potentially misleading for a reader who is only skimming the paper and has skipped to the discussion, as this difference in demographics between dog owners and non-dog owners is a major limitation. The purpose of my comment was not that we should necessarily look into age effects (as this is not the variable of interest here), but that it is an issue that the results are discussed in a way that doesn't highlight the other major difference between the two groups. In this large section on the first hypothesis, the difference in age distributions between the two groups should be clearly mentioned and it should be pointed out that as this analysis cannot determine causal effect and that, difference between the group of dog owners versus non dog owners could be related other factors such as age. While this is mentioned briefly in the limitations (lines 449-454) I would prefer to see this discussed as part of the discussion section on that specific hypothesis, as this is a specific limitation of that analysis and significantly impacts interpretability. If someone wanted to address this in future, it would need to be done using something like a causal inference approach.

Thank you for this valuable comment and additional explanation of what could be improved. In an effort to keep the structure of the Discussion section, with limitations all in one section and keeping the discussion of the limitation on our skewed age distribution there, we have in addition to stressing the skewed age (and gender) distribution in the result section now also highlighted this in the summary on outcomes with which we start the Discussion section as following (lines 403-408):

‘We found proof confirming our first hypothesis that dog owners do not recognise subtle dog pain signs more so than non-dog owners. However, we stress that our sample of dog owners and non-dog owners differed in demographic characteristics, such as age, as discussed below when we provide detail on the limitations of this study. Taking into account these demographic differences, dog owners achieved better results for identifying (…)’

Additional reviewer comments:

The author should clarify how they accounted for multiple comparisons. They mention a correction for multiple comparisons in the text, but it is not clear if it is applied correctly. Using the data they provided, I ran a chi-square using the author's methods (grouping neutral with unlikely) for the dog owner vs non-dog owner comparison. The proportion of owners/non-owners identifying the signs as potentially indicative of pain are the same as what they report, but the chi-square statistics and p-values I’m getting are different with no correction applied. This would indicate to me that a correction was applied, but it’s particularly confusing that they report a p-value for both freezing and turning head/body away that is larger than what I’ve found (this could be because they accounted for multiple comparisons), but the p-value they report for licking surfaces is smaller than the p-value I calculated. So, if they’re correcting for multiple comparisons, it’s not clear how this is done, since whatever they’re adjusting their p-values by should be the same for all rows of the table (i.e., all behaviours assessed)? P-values I calculated (with no correction for multiple comparisons for the three behaviours in question):

PS_freez: p = 0.00417; PS_turnaw: p = 0.00452; PS_licksurf = 0.04992. P-values reported in the paper (according to methods section, corrected for multiple comparisons): Freezing: p = 0.03; Turning head or body away: p = 0.03; Licking surfaces: p = 0.04. The authors reported in lines 187-189 that Bonferroni method adjusted p-values were used. However, from my understanding in that case the p-values should all have been adjusted using the same multiplier, which it appears they have not been. Could the authors please clarify how this correction was applied?

As we now present results with statistical testing via Mann-Whitney U tests, for which SPSS® does not include an option for Bonferonni correction, we have corrected manually, providing detail on the approach the Methods section lines 191-197.

---

## [Decision Letter · Decision Letter 3]

6 Feb 2026

Dear Dr. Herwijnen,

We look forward to receiving your revised manuscript.

Kind regards,

Carlos Alberto Antunes Viegas, DVM; MSc; PhD

Academic Editor

PLOS One

Journal Requirements:

Reviewers' comments:

Reviewer's Responses to Questions

**Comments to the Author**

Reviewer #1: All comments have been addressed

Reviewer #3: (No Response)

2. Is the manuscript technically sound, and do the data support the conclusions?

Reviewer #1: Partly

Reviewer #3: Yes

3. Has the statistical analysis been performed appropriately and rigorously?

Reviewer #1: Yes

Reviewer #3: Yes

4. Have the authors made all data underlying the findings in their manuscript fully available?

Reviewer #1: Yes

Reviewer #3: Yes

5. Is the manuscript presented in an intelligible fashion and written in standard English?

Reviewer #1: Yes

Reviewer #3: Yes

Reviewer #1: Thank you for responding to previous comments. There are just a few below that pertain to the updated analysis.

252 – “A significant difference was for the dog’s turning the head or body away “ re-word as doesn’t read well. Remove the dog’s

285 – remove dog’s from the sentences

Results: I am not sure it is correct to use “trend towards significance” as there is no accepted cut-off for this or definition. Maybe instead mention that it was not significant but is worth investigating further since the p value was low.

403 – I would not use the term “found proof” as it is not good practise to be proving hypotheses, maybe just say we confirmed our hypothesis.

420 - The discussion mentions that there was a significant difference between dog owners and non-dog owners in ascribing pain to licking surfaces. In the results it is not a significant after Bonferroni correction. The significance in this behaviour is between dog owners who have previously had a dog suffering from a painful condition compared to those who had not. This finding could be discussed here instead.

448 – the second sentence is long a difficult to follow. Maybe separate into two sentences for clarity.

Reviewer #3: Reviewer's Comments to Authors:

Title: The abilities in dog pain sign recognition as assessed by presenting seventeen listed dog behavioural signs and three case descriptions to dog owners and non-dog owners.

Recommendation: Minor revision

Overview and general recommendation

This manuscript covers a very interesting topic: the human’s abilities to recognise dog pain signs, comparing this ability between dog owners and non-dog owners and evaluating the influence possible pain experience. This topic is very important in term of dog welfare. Thereby, the paper covers a very noteworthy issue in the field of dog behaviour and welfare. The article is clearly laid out, and all the elements are present, referring to most relevant literature in the area, but according to me some part of the article need to be addressed: in particular the first part of the introduction need to be more focused on the topic. Fixing some issue, for me the paper could be accepted.

Specific comments are provided below.

Specific comments

Introduction

I found the first part of the introduction a little be out of focus and slightly forced. Please try to be more focused on your topic, especially the lines 40-55. It could be better discuss about pain, its characteristics, type and manifestations; but the part regarding aggression could be discussed in the Discussion section.

-Lines 63-65: this part should be moved in the Methods section.

-Lines 93-95: this sentence is note really clear, could you please rephrase it?

Methods

Lines 136-148: please add the description of the showed pain to led to a better link with the results and discussion.

Results

Lines 201-204: it should be noted and discussed in the discussion section that the non-dog owner sample was composed by person that never had a dog a person that lived with a dog in some point during their life. It is important because the living with a dog in the past may also have influenced their response.

It should be clarified and uniformed the characteristic of showed pain in the two of the three vignettes: it could be better defining the subtle and overt pain for the first and second case and maintain these terms consistently through the paper.

Discussion

Lines 407-410: for me this sentence is not clear could you please rephase it?

Lines 483-485: this sentence seems a little be disconnected from your results, please try to better connect these references (or otherwise remove them).

Lines 530-531: please try to link better this sentence.

**Do you want your identity to be public for this peer review?** For information about this choice, including consent withdrawal, please see our Privacy Policy

Reviewer #1: No

Reviewer #3: No

---

## [Author Response · Author response to Decision Letter 4]

8 Feb 2026

Response to each point raised:

Reviewer #1: Thank you for responding to previous comments. There are just a few below that pertain to the updated analysis.

Thank you again for your time and the opportunity to further improve our manuscript.

252 – “A significant difference was for the dog’s turning the head or body away “ re-word as doesn’t read well. Remove the dog’s

Thank you, reworded and removed.

285 – remove dog’s from the sentences

Thank you, removed.

Results: I am not sure it is correct to use “trend towards significance” as there is no accepted cut-off for this or definition. Maybe instead mention that it was not significant but is worth investigating further since the p value was low.

Thank you, ‘trend towards significance’ was removed, and we have detailed in the methods section which P-values we used as cut-off point (and why).

403 – I would not use the term “found proof” as it is not good practise to be proving hypotheses, maybe just say we confirmed our hypothesis.

The word ‘proof’ was removed.

420 - The discussion mentions that there was a significant difference between dog owners and non-dog owners in ascribing pain to licking surfaces. In the results it is not a significant after Bonferroni correction. The significance in this behaviour is between dog owners who have previously had a dog suffering from a painful condition compared to those who had not. This finding could be discussed here instead.

Thank you. We have removed the indication of this trend from the abstract and changed this part of the discussion.

448 – the second sentence is long a difficult to follow. Maybe separate into two sentences for clarity.

Thank you, this sentence was adapted.

Reviewer #3: This manuscript covers a very interesting topic: the human’s abilities to recognise dog pain signs, comparing this ability between dog owners and non-dog owners and evaluating the influence possible pain experience. This topic is very important in term of dog welfare. Thereby, the paper covers a very noteworthy issue in the field of dog behaviour and welfare. The article is clearly laid out, and all the elements are present, referring to most relevant literature in the area, but according to me some part of the article need to be addressed: in particular the first part of the introduction need to be more focused on the topic. Fixing some issue, for me the paper could be accepted.

Thank you for your kind words, review time and opportunity to improve our manuscript .

Introduction - I found the first part of the introduction a little be out of focus and slightly forced. Please try to be more focused on your topic, especially the lines 40-55. It could be better discuss about pain, its characteristics, type and manifestations; but the part regarding aggression could be discussed in the Discussion section. & Lines 63-65: this part should be moved in the Methods section.

Thank you, we have rewritten this section of the Introduction. After further consultation, also in light of previous review comments, we have kept the underpinning of study relevance intact. However, we hope to have rewritten the section to facilitate your comment too.

-Lines 93-95: this sentence is note really clear, could you please rephrase it?

We have rewritten these sentences.

Methods Lines 136-148: please add the description of the showed pain to led to a better link with the results and discussion.

We have added these pain descriptions.

Results Lines 201-204: it should be noted and discussed in the discussion section that the non-dog owner sample was composed by person that never had a dog a person that lived with a dog in some point during their life. It is important because the living with a dog in the past may also have influenced their response. It should be clarified and uniformed the characteristic of showed pain in the two of the three vignettes: it could be better defining the subtle and overt pain for the first and second case and maintain these terms consistently through the paper.

Thank you for your useful comment. We have checked for logical and consistent use of the terminology ‘subtle’ and ‘overt’ and made us take out these terms throughout the manuscript. We have however noted and discussed the point on the non-dog owner sample as following; ‘Also, our non-dog owning sample consisted of people presently not owning a dog. Consequently, previous experience with dogs could have affected our non-dog owning sample.’

Discussion Lines 407-410: for me this sentence is not clear could you please rephase it?

Thank you, this sentence was rephrased.

Lines 483-485: this sentence seems a little be disconnected from your results, please try to better connect these references (or otherwise remove them).

Thank you, this was removed.

Lines 530-531: please try to link better this sentence.

Thank you, this text was adapted.

---

## [Decision Letter · Decision Letter 4]

22 Feb 2026

The abilities in dog pain sign recognition as assessed by presenting seventeen listed dog behavioural signs and three case descriptions to dog owners and non-dog owners

PONE-D-25-09975R4

Dear Dr. Herwijnen,

We’re pleased to inform you that your manuscript has been judged scientifically suitable for publication and will be formally accepted for publication once it meets all outstanding technical requirements.

Kind regards,

Carlos Alberto Antunes Viegas, DVM; MSc; PhD

Academic Editor

PLOS One

Additional Editor Comments (optional):

Reviewers' comments:

Reviewer's Responses to Questions

**Comments to the Author**

Reviewer #1: All comments have been addressed

Reviewer #3: All comments have been addressed

2. Is the manuscript technically sound, and do the data support the conclusions?

Reviewer #1: Yes

Reviewer #3: Yes

3. Has the statistical analysis been performed appropriately and rigorously?

Reviewer #1: Yes

Reviewer #3: Yes

4. Have the authors made all data underlying the findings in their manuscript fully available?

Reviewer #1: Yes

Reviewer #3: Yes

5. Is the manuscript presented in an intelligible fashion and written in standard English?

Reviewer #1: Yes

Reviewer #3: Yes

Reviewer #1: All previous comments have been addressed. There are some convoluted sentences that may be better addressed by a proof reader.

Reviewer #3: (No Response)

**Do you want your identity to be public for this peer review?** For information about this choice, including consent withdrawal, please see our Privacy Policy

Reviewer #1: No

Reviewer #3: No

---

## [Editor Report · Acceptance letter]

PONE-D-25-09975R4

PLOS One

Dear Dr. Herwijnen,

I'm pleased to inform you that your manuscript has been deemed suitable for publication in PLOS One. Congratulations! Your manuscript is now being handed over to our production team.

Kind regards,

on behalf of

Dr. Carlos Alberto Antunes Viegas

Academic Editor

PLOS One